# A Pilot Experiment to Develop a Lightweight Non-Nuclear EMP Shelter Applying Civil-Military Cooperation in a Sustainability Policy

**Kukjoo Kim** [1,2], **Kyung-Ryeung Min** [3] **and Young-Jun Park** [1,2,*]

1   Department of Civil Engineering and Environmental Sciences, Korea Military Academy, Seoul 01805, Korea; klauskim@ufl.edu
2   Nuclear·WMD Protection Research Center, Korea Military Academy, Seoul 01805, Korea
3   ICT Polytech Institute of Korea, 16-26 Sunam-ro Gwangju, Gyeonggi 12777, Korea; iolapleader@gmail.com
*   Correspondence: parky@mnd.go.kr

**Abstract:** The goal of future wars is to incapacitate national core infrastructures through cyberattacks and electronic wars. The use of high-tech arms including high-power electronic weapons, laser weapons, and railguns to achieve a precise strike, minimum cost, and neutralization is gradually increasing. Considering the nuclear provocation and non-nuclear electromagnetic pulse (NNEMP) threats from North Korea, it has become urgent for Korea to expand its EMP protection systems. Hence, the need for developing a protective technology lighter than the conventional EMP protection technology is continuously being raised. However, no facility has applied such a lightweight protection technology thus far. Thus, this study tests the performance of a lightweight electromagnetic (EM) shielding material and evaluates the possibility of building a lightweight NNEMP shelter by installing the material. Among the commercially available EM shielding materials, only those appropriate for lightweight purpose are selected. Accordingly, the EM shielding performances of nine fabric types, five film types, and four wallpaper types are tested. For testing, a pan-type EM shielding room 2.5 m × 3.0 m × 2.5 m was constructed with a shielding performance of 80 dB at 18 GHz. The measurement method was based on the IEEE-STD-299 standard, and 10 frequencies from 14 kHz to 18 GHz were used. The result showed that the shielding performance was the highest in the 100 MHz band in most cases. In the high-frequency band above 1 GHz, the shielding performance was almost equal to, or slightly lower than, that in the 100 MHz band. This study confirms the feasibility of building lightweight NNEMP shelters in major military and civil facilities. If the NNEMP shelters to be constructed in military and civil facilities are replaced with lightweight shelters, approximately 49,862.4 tons of $CO_2$ emissions due to the concrete saved can be reduced per shelter. Assuming the Korean carbon transaction price to be USD 50/ton-$CO_2$, the saving amounts to US $2,493,120, contributing to the green growth policy of Korea.

**Keywords:** non-nuclear electromagnetic pulse (NNEMP); shielding effectiveness (SE); light weight EMP shelter; sustainable military policy; $CO_2$ emission

## 1. Introduction

### 1.1. Background

With the recent development of electronic devices and their increasing application, society has been exposed to an environment that is extremely vulnerable to high-power electromagnetic (EM) waves. Thus, the safety of major infrastructure facilities must be secured and systematic protection plans for realizing a response system to high-power EM attacks devised. Protecting the electronics

of major information and communication infrastructures from high-power electromagnetic pulses (EMP) is crucial to the national security [1,2]. However, the present EMP shelter construction methods and guidelines are based on the EMP protection technology and test standards of the US Department of Defense (e.g., MIL-STD-188-125). The EMP protection measures for the major target facilities in Korea are fairly poor considering that the EMP provocation by North Korea might result in serious social confusion [3,4]. Particularly, major infrastructures in the private sector must establish systematic protection measures considering the C4I system composition; however, no measures have been implemented owing to the lack of interest. For example, developed countries such as the US and European countries have committees that comprise experts from various fields, and technical research work is being conducted to establish EMP protection measures. However, Korea has not prepared any measures for critical facilities in the private sector despite the missile provocations by North Korea. The present social infrastructures are being operated in connection with electric, electronic, information and communication systems for the convenience of informatization and automation; however, in the event of any failures due to a high-power EMP, enormous damage to the entire social and national safety systems would result [5,6].

An instance of EMP-based military dangers was a nuclear explosion test with an explosive yield of 1.4 megatons conducted by the US 400 km above Johnston Island in the Pacific Ocean in 1962; this explosion knocked off the street and traffic lights and caused malfunctioning of the communication equipment in Honolulu, Hawaii, which was approximately 1400 km away from the test site. In July 1967, a missile mounted on a F-4 fighter on a USS Forrestal-class aircraft carrier, which was engaged in duty in the Vietnam War, was launched by an EM interference, causing 134 deaths and damage worth $75 million. Furthermore, during the Falkland War between the UK and Argentina in 1982, the UK battleships and Argentine fighters used radars of the same EM band. Consequently, continuous malfunctions occurred in the UK battleships, and an Argentine fighter failed to understand that there was a radar malfunction and crashed unprotected without taking any action. In 1980–1990, a UH-60 Blackhawk helicopter of the US Army crashed owing to the random activation of the safety device, and the investigation results revealed that the safety system had been exposed to a high-power EMP. Several similar helicopter crashes occurred subsequently, and the cause was always identified as a defect in the helicopter. As discussed in these examples, not only the high-altitude electromagnetic pulse (HEMP) caused by nuclear explosions but also the artificially generated high-power EMP have been very dangerous throughout the history of modern wars. Such non-nuclear EMPs (NNEMPs), which are also called high-power microwaves (HPM) or intentional EM interference, can damage electronic devices with a short pulse width, low repetition rate, and an output power ranging from hundreds of MW to several tens of GW. They are usually generated by electronic bombs (e-bombs) and previously appeared as military weapons; however, recently, they have been developed as compact, high-power devices. Products that can cause damage by generating a 350 MHz, attenuated vibration sine waves isotopically with an electric field strength of approximately 120 kv/m at 1 m and continuously activating five pulse waves for 30 min have been commercialized. Despite such threats associated with NNEMP, the present EMP protection measures of the military and national critical facilities are focused only on building protection facilities against HEMP [7–10]. Furthermore, it is estimated that enormous costs will be required to add the NNEMP protection ability to all the national infrastructures. The present EMP protection technology and test standards in Korea are based on MIL-STD-188-125, which is the EMP protection standard employed by the U.S. but is limited to only HEMP protection [8]. Figure 1 shows the classification of EMPs.

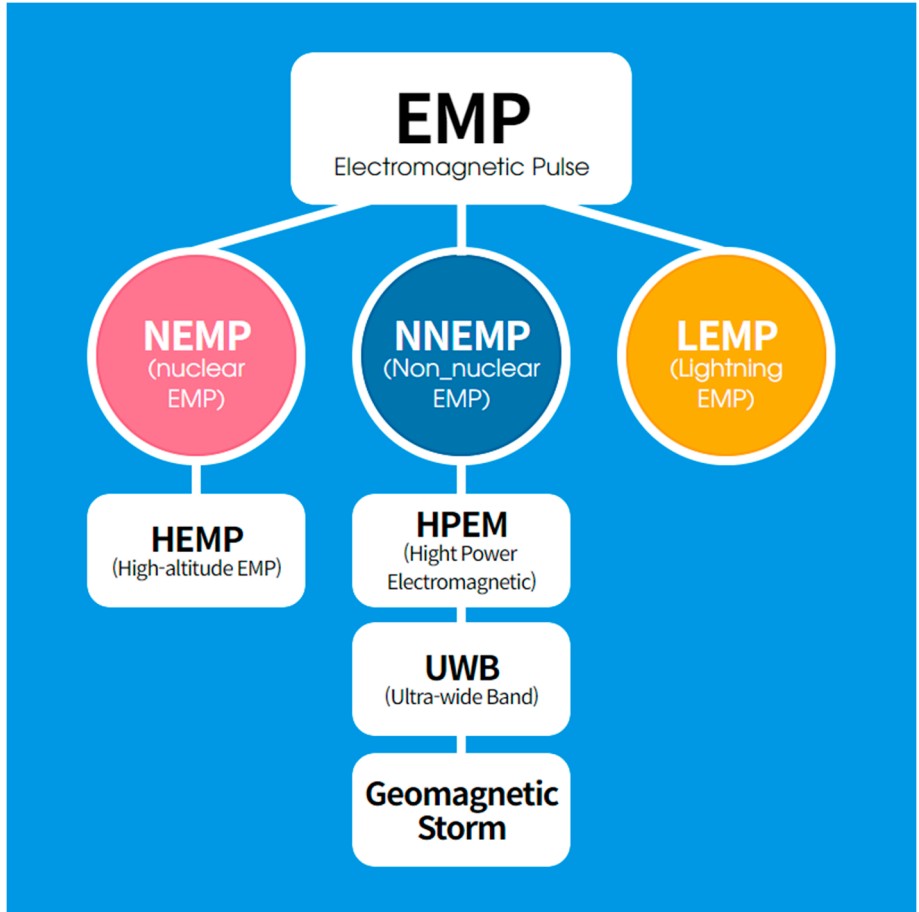

**Figure 1.** Classification of electromagnetic pulses (EMPs).

Non-nuclear, high-power EMP weapons are being developed in various forms including blaster guns, bombs, and radiofrequency-disturbance generating devices. The core technologies are miniaturization and low-voltage high-power generation. Ranets-E, which has been under development in Russia since 2008, can shoot down a fighter by using microwaves and might be deployed soon. In Korea, the Agency for Defense Development has been actively researching EMPs, not only for EMP protection but also for realizing EMP-based attack weapons. The present EMP protection method is to shield the target to be protected using a metal enclosure, and the objects of protection include major C4I (i.e., command, control, communication, computer, and intelligence) facilities and communication equipment of the national critical facilities. To secure the existing facilities and specially constructed protective facilities, large amounts of concrete, reinforcing bars, and steel plates should be used. Because the resulting costs would be high, the difficulty in securing the required budget has delayed the construction of such protective facilities. Moreover, reducing the concrete and steel usage in construction projects is crucial for achieving sustainability awareness and green planning [11]. According to the International Energy Agency and United Nations Environment Programme, building construction and operations account for approximately 36% of the global energy consumption. This corresponds to 40% of the global energy-related carbon dioxide ($CO_2$) emissions in 2017; these emissions must be decreased to prevent the global climate change [12]. Pacheco-Torgal et al. found that the use of concrete and steel bars accounted for 65% of building greenhouse gas emissions and the use of concrete for 40% of $CO_2$ emissions [13]. Particularly, the mean embodied carbon dioxide of a building is 340 kg-$CO_2$/m$^2$, which accounts for approximately 60% of the total structure [14]. This suggests that reducing the concrete and steel bar usage in construction projects is critical to reducing the carbon emissions [15–19]. The sustainability policy of the military forces also needs to adopt the principle of green growth [20].

Therefore, as part of NNEMP protection measures, we aim to conduct an exploratory research on the measures required to construct lightweight protective facilities that can be applied to the private and military sectors. To that end, the possibility of developing a lightweight NNEMP shelter through verification experiments on lightweight EMP shielding materials is explored. The materials demonstrated in this study might be used for constructing NNEMP protection facilities in a quick and flexible manner in the private and military sectors.

*1.2. Objectives and Scope*

This study mainly aims to comparatively verify different NNEMP shielding materials applicable to private and military facilities. To that end, lightweight EMP shielding materials were selected, and their EMP shielding performances were tested. Through EMP shielding tests, we measured the performances of materials in the frequency band of 9 kHz–18 GHz using an IEEE-STD-299-based measurement method, which could test the high-frequency region of NNEMP. From the test results, we confirmed the possibility of developing lightweight NNEMP shelters in future, and the associated economic effects were considered by $CO_2$ emission and cost reduction analysis.

## 2. NNEMP Lethality

The realization that high-power electromagnetic waves (HPEM) are destructive and can damage electronic devices first emerged in the early 1970s. Consequently, the development of HPEM sources began. Traditional microwave sources such as radars are implausible as an HPEM source because their pulse powers do not exceed 100 MW. However, there is no generally accepted definition. The generation of and protection from HPEM has been a long researched subject, and the US and Russia are striving to research and develop strong HPEM sources. Many countries have been performing a wide range of activities regarding HPEM protection devices. The main objective of an HPEM weapon is to disturb and damage the functions of systems that depend on electronic devices. Even a considerably short disturbance can cause delay defects that could restart vehicles and computers. Serious problems can occur if these defects occur in airplanes or robots. Disturbances can occur in unprotected electronic devices in low electric field strengths more than disturbances for devices that are close to the transmitted mobile phone.

Because the operations of major national infrastructures (e.g., power grids, communication networks, financial networks) are being accelerated and automated, the application range of electronic devices is continuously widening. Furthermore, electric and electronic devices have become gradually smaller and more sensitive than before. The development of strong EMP sources with the direct goal of incapacitating the electronic devices of military, public, and private systems is on a gradual rise. If the energy received by electronic devices is sufficiently high to melt or destroy their circuit boards or semiconductor materials, permanent damages might occur. HPEM radiation can comprise simple pulses or pulse bursts (pulse trains). If inflicting permanent damage is the aim, it may be wise to concentrate the energy on a small number of pulses. However, to cause only a dysfunction, it might be necessary to exceed a specific pulse repetition frequency.

A strong HPEM source can cause serious damage. This type of weapon is usually employed when the level of conflict between two warring parties is high, as it requires a large amount of materials, economic resources, and information. Some HPEM weapons known as low-frequency pulse weapons have properties appropriate for causing destruction and performing terrorist attacks. These weapons can be used easily because they do not require a high level of knowledge to produce the components. If the intended use is destruction, HPEM weapons may be used to generate HPEM radiation effects to destroy traditional microwave sources and electronic systems. As an information war strategy, HPEM weapons might be used in the early stage of a battle to incapacitate the opponent. Figure 2 depicts the destruction of a major C4I system using an e-bomb.

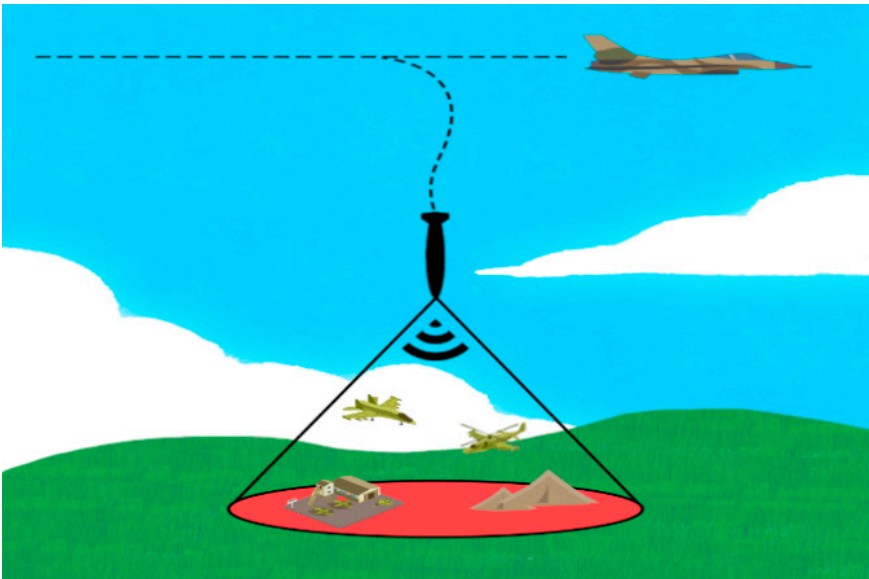

**Figure 2.** Lethal footprint of an high-power microwaves (HPM) e-bomb [21].

In future, the performance of HPEM weapons will be developed towards achieving power increase, pulse repetition frequency, and antenna gain, and intelligence will be provided in the form of signal modulation in a manner similar to that in telecommunication war. The advantage of HPEM weapons is that the radiation arrives at the target at the speed of light, and these weapons can attack multiple targets even without any detailed prior information about the target system. Furthermore, HPEM weapons can attack sophisticated (electronic-device dependent) targets with simple resources under all weather conditions. However, one disadvantage of HPEM weapons is that they can be used only for electronic systems. Additionally, the firing range is limited, and the HPEM source can be located by detecting its radiation. If the HPEM possesses an electric field strength of several hundred V/m, it can cause disturbances in unshielded systems. Generally, an electric field strength higher than approximately 10 kV/m is required to cause destruction. Highly powerful HPEM weapons can destroy a target at a 1 km distance provided there are no obstacles in the middle. A device developed by DIEHL is shown in Figure 3, and it can be used by the military or police to incapacitate the communication equipment of the opponent [22].

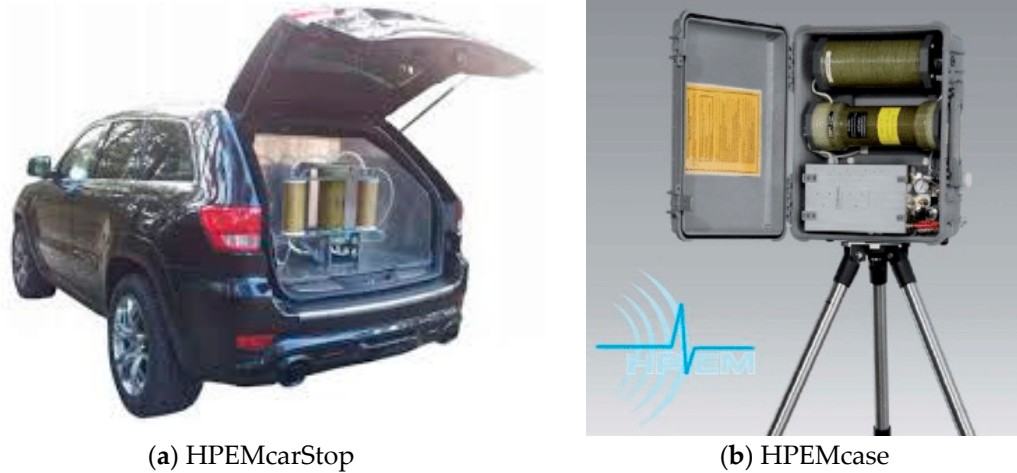

(**a**) HPEMcarStop        (**b**) HPEMcase

**Figure 3.** Commercialized high-power electromagnetic waves (HPEM) system [22].

The term "front-door combination" is used when a radiation penetrates through an opening of antennas or sensors designed to receive EM radiations. If the working frequency of this sub-system is identical to that of HPEM radiation, the first type of front-door combination is used; otherwise, the second type is used. An example of the first type of front-door combination is to hit a wireless link using HPEM (when the HPEM frequency lies in the frequency band of the wireless link). The term "back-door combination" is used when the radiation (after penetrating a weak point that always exists in the shield) is combined with a conducting wire and cable, and then with an electronic component. For an unshielded target, the radiation is directly combined with the conducting wire and cable. The imperfect parts of the shield include a non-conductive gasket, screw fastening, rivet joint, vent, drain, and display window. To protect electronic devices by back-door combination, the electronic device must be installed in a well-shielded space, and all the cables and conducting wires connected to the electronic device must be shielded and filtered in a reliable manner. This type of requirement can be reduced depending on the electronic device location. Concrete, dust, and rocks provide a certain amount of shielding effect. Regardless of whether the device has a metal case or a thick concrete wall, the shield must be free of large openings. Figure 4 depicts the ranges of national systems that are vulnerable to EMP attacks [21].

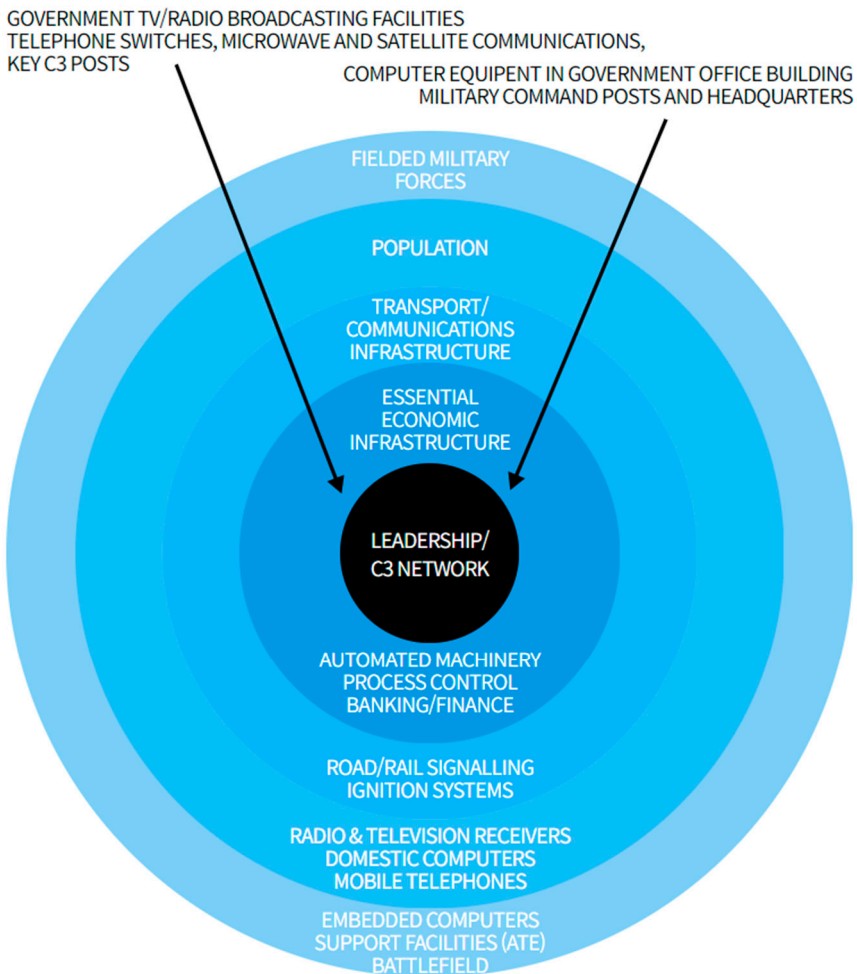

**Figure 4.** Electromagnetically vulnerable target sets [21].

## 3. Experimental Details

### 3.1. Test Specimens

With the increasing diversity of methods that employ EM waves and the widening applications of electric and electronic devices that use EM waves, the exposure to the EM energies of various frequency bands is rising. Furthermore, many countries are making efforts to prevent the dangers associated with EM waves through electromagnetic compatibility—an international standard for EM interferences. The effective prevention of EM interferences is not only a key requirement for improving the performance and extending the life of electronic components, but also a critical factor for improving the quality of life by reducing the adverse effects of harmful EM waves on the human body. Therefore, all the sources that generate EM energy must be shielded. EM shielding has been emerging as a measure to prevent such harmful EM interferences. For EM shielding, the characteristics of the shielding effect are considered important irrespective of the structure and shape of the shielding material. A high EM shielding effect is related to a satisfactory protection level. For example, many metals with appropriate electric properties are used as EM shielding materials; however, using metals introduces problems of heaviness, easy corrosion, and poor processability. To ameliorate these disadvantages, lightweight metal–polymer composites with environmental resistance and high productivity are drawing attention. Fiber materials and structures with EM properties can be used for EM shielding. Fiber materials have an excellent electrical insulation property. Most fibers are fabricated using natural or synthesized polymers that have a high electrical resistance. However, the electrical conductivity of fiber materials is required in specific applications including electric heating, EM wave protection, and signal transmission. Therefore, many attempts are being made to produce conductive fibers. The currently commercialized EM shielding materials related to construction can be broadly classified into four types: fabrics, sheets, wallpapers, and paint. They also include the new materials under development. These materials are mixed with other metal components to maximize the shielding performance against EM waves, thereby enhancing the durability, corrosion resistance, and EM shielding performance. The characteristics of each type of the abovementioned EM shielding materials are discussed as follows.

First, conductive fibers have excellent EM shielding properties and offer many advantages such as a lower weight than metal foils or grids, elasticity, porosity, air permeability, and corrosion resistance. They are produced using various methods through mixing with metals. An example is coating the outer part of a polyester fabric yarn with metal components such as silver (Ag) and copper (Cu) to impart EM shielding.

Second, an EM shielding sheet is fabricated by stacking thin layers of constant thickness and then applying a conductive adhesive to attach the layers. EM shielding sheets are becoming thinner, more flexible, and more durable with technical development.

Third, EM shielding paint, which is usually produced by dispersing metal fillers (such as nickel, copper, and silver) on a surface composed of resins (such as acrylic and urethane polymers), offers many advantages. It does not require any new investment because the existing production line can be used without change, and products can be easily spray painted. It can be applied to each plastic type by spraying it over the regular paint coating for reinforcing the surface of a foundation system. Furthermore, it is very cheap compared with other EM shielding methods. However, EM shielding paint has many disadvantages. It cannot be finished owing to its low resistance compared with plating and deposition methods. Additionally, the film thickness is high at approximately 45 $\mu$m. Hence, the film peels off, and the coating thickness on the surface remains non-uniform unless a robot sprayer is used. Furthermore, a long-term shielding effect cannot be achieved owing to the oxidation of fillers such as copper.

To conduct this study, we could not purchase and test all the EM shielding materials available on the national and international markets owing to the time and cost constraints. To address this physical limitation, only the products appropriate for the experiment were selected in advance (see Table 1).

**Table 1.** Test specimens used for the shielding effectiveness (SE) test.

| Classification | Manufacturer | Product Number |
|---|---|---|
| Shielding fabrics (nine types) | Samgang tech | SGF-D130 |
| | Samgang tech | SGF-D150 |
| | Samgang tech | SGF-WD270 |
| | A-Jin Electron | W-290-PCN |
| | Holland Shielding | Systems BV 4711 series |
| | Less EMF Inc. | COBALTEX |
| | Less EMF Inc. | NICKEL/COPPER RIPSTOP FABRIC |
| | Less EMF Inc. | PURE COPPER POLYESTER TAFFETA |
| | Less EMF Inc. | SILVER MESH FABRIC |
| Shielding wallpapers (four types) | Hana Elecom | CFT-235-FR-NH |
| | Hana Elecom | CFT-290-FR-NH |
| | Less EMF Inc. | Stick E Shield |
| | Y-Shield | YCF-60-100 |
| Shielding films (five types) | EMCPRO | SF2209 |
| | Whil KOR | WT 70 MNT |
| | ShieldGreen | SGWF26 |
| | Less EMF Inc. | Scotch Tint |
| | Less EMF Inc. | Scotch Tint Super |

*3.2. Experimental Procedures*

To test the EM shielding materials, a pan-type EM shielding room 3 m × 2.5 m × 2.5 m was constructed, and an opening was made on 1 side of the room to measure the shielding effect of the material. The tests were conducted on over 10 representative frequency bands using the method of IEEE-STD-299. The shielding performance of the EM shielding room was designed to be 80 dB at 18 GHz. To study the shielding effects of various EM materials, a 60 cm × 60 cm entry plate was mounted on the walls of the shielding room. The size of the opening on the wall was selected in accordance with MIL-DTL-83528G, a detailed standard of the US Department of Defense. The bolt spacing of the entry plate was designed to be 5 cm or less according to the standard. The bolt spacing in the edges was designed to be 2.5 cm or less to prevent degradation of the shielding effect. The height of the center of the entry plate was designed to be 1.25 m to fix the antenna height to 1.25 m (center of the shielding room). To enable various tests according to the material, pan-type, glass, acrylic, and fiber connection frames were installed, and the frames were attached to glass or acrylic during the shielding film test, following which the shielding performance was measured. Figure 5 depicts the shielding room designed for performing the shielding effectiveness (SE) test.

According to IEEE-STD–299, tests must be performed at locations where the likelihood of EM wave leakage in the 9 kHz–20 MHz band is high and at fixed intervals along horizontal and vertical directions in the frequency band of 20 MHz to 18 GHz. The IEEE-STD-299 standard is for testing the shielding performance of an EM shielding enclosure in the frequency range of 9 kHz–18 GHz (expandable to 50 Hz to 100 GHz). Generally, the minimum length of each side of a shielding enclosure is 2 m. The shielding performance is the difference in strength between the signal received without the shielding enclosure and that received with the shielding enclosure. Furthermore, the shielding performance is expressed in decibel (dB) and calculated as the ratio of the reference signal level recorded in the calibration procedure to the measure signal level. A signal generator can be embedded in the receiver or spectrum analyzer. Radio frequency signals can be transmitted through the shield using an appropriate fiber optic line or a high-quality, metallic shielding coaxial cable. In the low-frequency or magnetic range (10 kHz–20 MHz), an electrostatically shielded annular antenna is required. In the resonance frequency range (20 MHz–approximately 300 MHz), a biconical antenna must be used. In the high-frequency or plane wave range (approximately 300 MHz–1 GHz), a log-periodic antenna or an equivalent broadband antenna must be used. The transmission and reception of the antenna must

be performed in the specified frequency band, and the operating area requirements must be satisfied with regard to other test equipment.

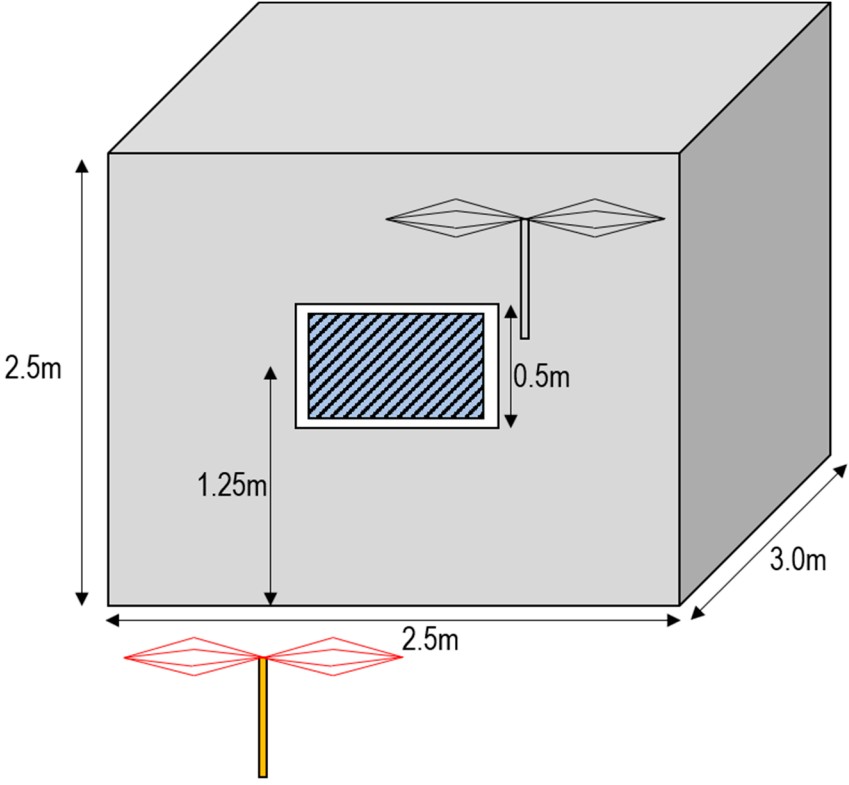

**Figure 5.** SE test setup.

The measurement frequencies were selected from the representative frequencies suggested in IEEE-299.1. For each frequency band and antenna type, two to four frequencies were selected. From the band of the loop antenna, 14 and 200 kHz were selected. From the band of the biconical antenna, 50 and 100 MHz were selected. From the band of the log-periodic antenna, 400 MHz and 1 GHz were selected. From the band of the horn antenna, 3, 6, 10, and 18 GHz were selected as frequencies that could represent each antenna type. Considering the characteristics of the surrounding environment noise and reception system, the measurement setup was configured differently for the loop antenna band and for the remaining antenna band. The shielding performance was calculated as the ratio of the measured signal to the reference signal. To derive highly accurate data, even the noise level in the measurement environment was considered. For measuring the reference signal level, two antennas were placed opposite each other at a distance suggested in the standard, and a certain level of signals was transmitted. Subsequently, the received level was recorded as the reference signal. For measuring the noise level in the measurement environment, all the transmission devices were powered off in the same setup, and the value of the received signal in each frequency band was recorded as the noise level. To minimize the environmental error, the environmental noise reference value was measured at a space 2 m away from the shielding room. The measured environmental noises were similar in the range of 90–100 dB for all the frequency bands. The measured reference value underwent the highest loss of 33.5 dB in the 14 kHz band for a distance of 60 cm between the transmission and reception antennas. This loss was attributed to the magnetic field characteristics, which had the highest loss rate by distance. To compensate for this loss, an annular reception antenna equipped with a signal amplifier was used for the measurements. Furthermore, considering the frequency characteristic which had a high signal loss for the cable in the several GHz frequency band, short cables 3 m in length were used for connection to the transmission and reception antennas. Based on the abovementioned

measurement results, the difference between the reference value and environmental noise became the operating range according to the current equipment configuration and environment. The EM shielding room had the minimum shielding performance of 71.2 dB in the lowest frequency band of 14 kHz and the maximum shielding performance of 113.3 dB in the 400 MHz band. In conclusion, the EM shielding room could be used to test the shielding performances of the materials.

### 3.3. Shielding Effectiveness: Test Results

Considerable attention has to be paid to the installation method when the shielding performance is tested after installing a material in the shielding room. Accordingly, it was observed that the actual material installation method significantly affected the performance measurement results of the materials. Consequently, in the case of wallpaper, to enhance the electric conductivity of the contact surface between the material and shielding room, the part fixed to the shielding room was attached using a conductive adhesive, and then the material was fixed and measured. In the case of film, to enhance the electric conductivity between the film and shielding room wall after attaching the film to an acrylic fixing frame, a foam gasket was attached inside the acrylic fixing frame before measurements. Furthermore, to prevent the measurement result from being influenced by the inconstant fastening force of the bolts when the frame of a material was fixed, the material was fixed with a certain amount of force using a torque wrench.

The shielding performances were measured according to IEEE-STD-299 after the materials were installed on the shielding room wall. The results showed that among the films, the SGWF26 model of Shield Green showed the highest shielding performance of 48.0 dB in the 100 MHz frequency band and generally performed well in the other frequency bands as well. Among the fiber materials, the SGF-D130 model of Samgang New Materials Co., Ltd., a Korean company, showed the highest shielding performance of 95.3 dB in the 100 MHz frequency band. Among the wallpapers, the CFT-290-FR-NH model of Hana Elecom showed the highest shielding performance of 92.3 dB in the 100 MHz frequency band. Figure 6 shows the test setup, including the shielding room, for measuring the shielding performance of lightweight shielding materials.

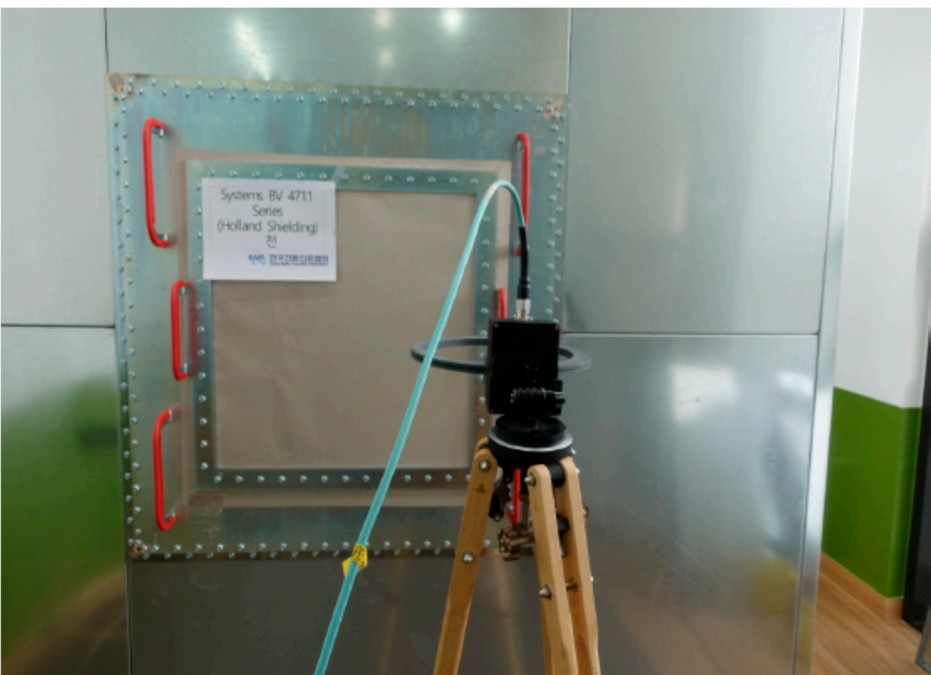

**Figure 6.** SE test setup.

Table 2 lists the EMP shielding efficiencies of various products in each frequency band.

**Table 2.** SE test results of the specimens.

| Product Number | Antenna | 14 kHz | 200 kHz | 50 MHz | 100 MHz | 400 MHz | 1 GHz | 3 GHz | 6 GHz | 10 GHz | 18 GHz |
|---|---|---|---|---|---|---|---|---|---|---|---|
| | | | | | | Frequency | | | | | |
| SGF-D130 | Horizontal | 5.8 | 19.5 | 88.3 | 95.3 | 80.3 | 76.5 | 78.9 | 76.1 | 75.2 | 69.3 |
| | Vertical | 7.0 | 23.3 | 90.9 | 92.6 | 87.6 | 80.3 | 73.6 | 71.0 | 73.7 | 70.7 |
| SGF-D150 | Horizontal | 5.7 | 10.6 | 59.4 | 67.6 | 62.1 | 58.7 | 73.1 | 68.6 | 70.2 | 68.4 |
| | Vertical | 4.9 | 15.9 | 63.9 | 71.6 | 72.8 | 63.5 | 53.1 | 58.7 | 64.1 | 64.0 |
| SGF-WD270 | Horizontal | 6.2 | 18.5 | 69.4 | 78.6 | 75.1 | 73.8 | 74.1 | 69.8 | 69.3 | 65.9 |
| | Vertical | 6.6 | 20.5 | 70.1 | 77.1 | 80.2 | 71.3 | 71.4 | 73.8 | 65.4 | 72.0 |
| W-290-PCN | Horizontal | 5.5 | 16.2 | 64.9 | 73.3 | 69.6 | 63.1 | 79.5 | 71.2 | 69.1 | 64.9 |
| | Vertical | 6.9 | 23.5 | 73.8 | 81.2 | 82.9 | 72.2 | 63.0 | 64.6 | 62.3 | 71.1 |
| Systems BV 4711 series | Horizontal | 7.3 | 24.2 | 73.9 | 81.9 | 78 | 80 | 66.7 | 51.8 | 57.4 | 66.8 |
| | Vertical | 6.1 | 20.1 | 70.7 | 76.3 | 77.4 | 67.3 | 62.9 | 56.9 | 55.4 | 67.3 |
| COBALTEX | Horizontal | 5.6 | 19.9 | 74.2 | 82.8 | 82 | 84.3 | 81.2 | 86.6 | 78.4 | 71.9 |
| | Vertical | 6.1 | 22.2 | 70.3 | 77.3 | 77.7 | 72 | 84 | 90.4 | 73.3 | 73.3 |
| NICKEL/COPPER RIPSTOP FABRIC | Horizontal | 6.3 | 23.2 | 74.8 | 83 | 79.1 | 76.7 | 78.3 | 78.6 | 75.8 | 69 |
| | Vertical | 7.5 | 25.3 | 76.7 | 84.8 | 85.5 | 78.5 | 66.3 | 73.4 | 70.3 | 73 |
| PURE COPPER POLYESTER TAFFETA | Horizontal | 7.8 | 27.1 | 74.4 | 84.3 | 84 | 79.1 | 74.3 | 69.3 | 66.2 | 66.8 |
| | Vertical | 7.4 | 26 | 78.7 | 83.8 | 80.7 | 67.3 | 75 | 74.6 | 63 | 74.9 |
| SILVER MESH FABRIC | Horizontal | 3 | 3.5 | 64.5 | 66.6 | 50.4 | 37.6 | 41.6 | 39.7 | 32.5 | 34.4 |
| | Vertical | 2.7 | 1.2 | 64.4 | 65.1 | 50.2 | 34.9 | 36.6 | 39.8 | 30.6 | 38.6 |
| CFT-235-FR-NH | Horizontal | 6.1 | 17.8 | 75.6 | 80.5 | 74.3 | 67.2 | 64.5 | 48.2 | 54.5 | 65.2 |
| | Vertical | 6.3 | 19.0 | 78.6 | 86.5 | 74.1 | 67.2 | 68.4 | 61.1 | 52.4 | 67.2 |
| CFT-290-FR-NH | Horizontal | 5.0 | 13.2 | 84.6 | 88.7 | 73.3 | 57.3 | 72.3 | 66.0 | 68.2 | 66.9 |
| | Vertical | 6.3 | 21.1 | 88.9 | 92.3 | 83.5 | 69.7 | 58.2 | 67.2 | 60.2 | 65.5 |
| Stick E Shield | Horizontal | 6.5 | 25.0 | 79.7 | 88.4 | 86.3 | 82.8 | 69.3 | 70.2 | 59.4 | 61.8 |
| | Vertical | 6.5 | 23.4 | 76.0 | 86.0 | 83.6 | 74.3 | 69.4 | 73.3 | 57.0 | 72.5 |
| YCF-60-100 | Horizontal | 6.7 | 22.4 | 71.6 | 79.0 | 72.4 | 62.5 | 54.7 | 46.5 | 49.7 | 51.3 |
| | Vertical | 6.8 | 22.1 | 72.6 | 76.6 | 71.7 | 55.6 | 52.1 | 49.9 | 51.0 | 53.2 |
| SF2209 | Horizontal | 2.0 | 2.9 | 27.8 | 38.9 | 38.0 | 35.6 | 34.2 | 32.6 | 34.4 | 37.5 |
| | Vertical | 2.1 | 3.1 | 27.5 | 37.2 | 37.9 | 30.1 | 31.6 | 40.6 | 32.1 | 40.7 |
| WT 70 MNT | Horizontal | 1.8 | 2.5 | 19.9 | 27.6 | 26.7 | 27.0 | 19.8 | 20.0 | 19.9 | 22.7 |
| | Vertical | 1.8 | 2.5 | 18.9 | 25.1 | 25.0 | 17.2 | 15.5 | 22.2 | 17.0 | 32.2 |
| SGWF26 | Horizontal | 2.2 | 2.9 | 43.8 | 48.0 | 25.1 | 24.1 | 20.2 | 23.6 | 21.0 | 27.0 |
| | Vertical | 2.4 | 3.1 | 44.9 | 46.1 | 25.3 | 22.5 | 18.2 | 26.9 | 19.8 | 30.2 |
| Scotch Tint | Horizontal | 3.9 | 6.0 | 18.2 | 25.2 | 22.6 | 24.5 | 18.5 | 21.6 | 21.6 | 26.2 |
| | Vertical | 5.4 | 6.0 | 18.4 | 24.8 | 24.1 | 20.7 | 18.4 | 21.6 | 20.4 | 30.8 |
| Scotch Tint Super | Horizontal | 1.0 | 1.6 | 22.3 | 30.9 | 29.2 | 26.8 | 22.5 | 30.9 | 28.6 | 30.9 |
| | Vertical | 1.5 | 1.6 | 26.3 | 36.3 | 31.2 | 27.2 | 25.0 | 31.8 | 24.5 | 38.3 |

## 4. Conclusions

This study tested the performances of several lightweight EMP shielding materials, following which conclusions about a more efficient EMP protection technology were drawn through actual installation and performance measurement. Based on the results of this study, we plan to apply a lightweight EMP protection technology to major national infrastructures. To achieve this goal, we selected the appropriate EMP shielding material and then tested and verified it.

To select the appropriate shielding material, the technical trends of EMP shielding performance measurement methods and materials were analyzed using various national (Korean) and international books and studies. Based on the trend data derived, among the various commercial products on the market, only the EMP shielding materials appropriate for lightweight purposes were investigated and listed. For the material types, we tested fabrics, films, wallpapers, and paints, all of which were appropriate for lightweight purposes. However, because we could not purchase and test all the investigated EMP shielding materials owing to time and cost limitations, only the products appropriate for actual tests were selected using some specific criteria. According to these selection criteria, nine fabric types, five film types, and four wall paper types were selected. The materials un-related to EMP shielding or whose performance was difficult to measure were excluded from the test.

To measure the EMP shielding performance by actual material installation, a pan-type EMP shielding room 2.5 m × 3.0 m × 2.5 m was constructed with a shielding performance of 80 dB at 18 GHz. The shielding effect of the materials was measured by making an opening on one side of the shielding room. The measurement was performed following the method in IEEE-STD-299. Ten frequencies between 14 kHz and 18 GHz were used for the measurements. The shielding performance was measured with horizontal and vertical biases for each frequency. Summarizing the measurement results of the derived materials, the shielding performance was the best in the 100 MHz band in most

cases. The shielding effectiveness in the high frequency band above 1 GHz was similar to or lower than that at 100 MHz, and the shielding performance of some materials substantially deteriorated.

We confirmed the possibility of building lightweight NNEMP shelters in private and military facilities using lightweight shielding materials. The application of lightweight shielding materials to major military and private facilities will be also conform to the civil–military sustainability policy. Furthermore, replacing many EMP shelters to be constructed in future with lightweight protection facilities will not only save a large amount of concrete but also offer many advantages in terms of reduced $CO_2$ emissions for the concrete saved. For example, assuming the unit $CO_2$ emission of ready-mixed concrete to be 3.152 ton-$CO_2$/ton [23], approximately 49,862.4 tons of $CO_2$ emissions generated by the construction of one EMP shelter can be saved. Assuming the Korean carbon transaction price to be USD 50/ton-$CO_2$ [24], the saving of US $ 2,493,120. Lightweight NNEMP shelters could be highly useful in military facilities that need to be transported between operations. Furthermore, in the civil sector as well, rather than installing and operating the highly expensive EMP protection facilities in peace time, lightweight NNEMP shelters can be flexibly installed in case of a crisis. Lightweight NNEMP shelters can also contribute to the green growth policy of Korea.

From the performance measurement results of the EMP shielding materials studied, it was evident that although the materials could not reach the shielding performance of steel plates, which are widely used in construction, some materials exhibited excellent shielding performances in a wide frequency band. However, from the viewpoint of nuclear or non-nuclear EMP protection, it was difficult to sufficiently satisfy the performance requirements of EMP shelters by using only one shielding material. Therefore, we require measures to improve the shielding performance and material durability in various ways, such as combining diverse shielding materials or stacking materials. To improve the shielding performance more effectively, further research is required into installing different EMP shielding materials in a comparative experiment in experimental facilities similar to the actual environment of general offices.

**Author Contributions:** Conceptualization, Y.-J.P. and K.K.; Methodology, K.K. and K.-R.M.; Software, K.-R.M.; Validation, Y.-J.P., K.K. and K.-R.M.; Formal Analysis, K.K.; Investigation, K.-R.M.; Resources, Y.-J.P.; Data Curation, K.-R.M. and K.K.; Writing-Original Draft Preparation, K.K.; Writing-Review & Editing, Y.-J.P.; Visualization, K.-R.M.; Supervision, Y.-J.P.; Project Administration, K.-R.M. and Y.-J.P.; Funding Acquisition, Y.-J.P. All authors have read and agreed to the published version of the manuscript.

**Funding:** This research was supported by a grant (20SCIP-B146646-03) from the Korea Agency for Infrastructure Technology Advancement.

**Acknowledgments:** This work was supported by the research fund of the Korea Agency for Infrastructure Technology Advancement. The ROKA Nuclear·WMD Protection Research Center at Korea Military Academy is gratefully acknowledged for providing their support.

**Conflicts of Interest:** The authors declare no conflict of interest. The funders had no role in the design of this study, collection, analyses, or interpretation of the data, writing of the manuscript, or in the decision to publish the results.

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
