# Peer review of "A Pilot Experiment to Develop a Lightweight Non-Nuclear EMP Shelter Applying Civil-Military Cooperation in a Sustainability Policy"

_sustainability, doi:10.3390/su122410669_

Round 1
Reviewer 1 Report
- The authors worked on a useful and timely topic, regarding a sustainability policy for the Military-Civil cooperation. In detail, the authors measured the capability of various EMP shielding materials against a non-nuclear EMP environment. This reviewer agrees that EMP threat, especially non-nuclear EMP, is a serious threat for the military and government.
- Overall, the manuscript is structured well, on the aspect of content and context. The scope of this study falls within the subject matter of Sustainability, as the authors highlighted that a significant reduction in concrete use could be achieved by using thinner/lighter materials.
- However, this reviewer has a major concern about the aspect of sustainability addressed in this paper. The authors underlined the benefits of reduced concrete amounts, in line with cost savings. However, additional costs associated with potentially using more expensive materials were not clearly addressed. More specifically, it would be great if the authors discuss further information about the selected shielding materials, such as their additional costs, emissions, sustainability impacts.
Author Response
First, the authors would like to acknowledge and thank all reviewers for providing invaluable and constructive review comments in relation to this article. These comments are sincerely appreciated and have clearly made a positive impact on the quality of the paper. Based on the reviewers’ suggestions, changes were done to the paper to make it hopefully clearer and more understandable.
- The authors worked on a useful and timely topic, regarding a sustainability policy for the Military-Civil cooperation. In detail, the authors measured the capability of various EMP shielding materials against a non-nuclear EMP environment. This reviewer agrees that EMP threat, especially non-nuclear EMP, is a serious threat for the military and government. Overall, the manuscript is structured well, on the aspect of content and context. The scope of this study falls within the subject matter of Sustainability, as the authors highlighted that a significant reduction in concrete use could be achieved by using thinner/lighter materials. However, this reviewer has a major concern about the aspect of sustainability addressed in this paper. The authors underlined the benefits of reduced concrete amounts, in line with cost savings. However, additional costs associated with potentially using more expensive materials were not clearly addressed. More specifically, it would be great if the authors discuss further information about the selected shielding materials, such as their additional costs, emissions, sustainability impacts.
Re: The authors appreciate your valuable comment. EMP protection in the current military uniformly requires 80dB of shielding effectiveness. This is the first study to evaluate new materials with differential shielding effectiveness in the military. The new shielding materials are made from carbon and steel fibers and evaluating the impact of these materials in term of a sustainability perspective will be a whole new study. The main objective of this study is to determine whether it is possible to build a lightweight EMP protection facility with differential shielding effectiveness. In addition, the authors believe that EMP protection using new materials is a very important matter from the viewpoint of military sustainability.
Reviewer 2 Report
Thank you for the opportunity to review your paper. I agree that EMPs present a significant future threat for militaries and governments around the world.
This paper is very well organized, and easy to follow. The research methods are clearly described.
The results and conclusions support the research approach.
This research only claim the benefits of reduced concrete, including tons, emissions, and emission cost-avoidance.
If you analyze the additional cost and then evaluate the economic benefits including costs, emissions, sustainability impacts, etc., it will be a better article.
Author Response
First, the authors would like to acknowledge and thank all reviewers for providing invaluable and constructive review comments in relation to this article. These comments are sincerely appreciated and have clearly made a positive impact on the quality of the paper. Based on the reviewers’ suggestions, changes were done to the paper to make it hopefully clearer and more understandable.
- Thank you for the opportunity to review your paper. I agree that EMPs present a significant future threat for militaries and governments around the world. This paper is very well organized, and easy to follow. The research methods are clearly described. The results and conclusions support the research approach. This research only claim the benefits of reduced concrete, including tons, emissions, and emission cost-avoidance. If you analyze the additional cost and then evaluate the economic benefits including costs, emissions, sustainability impacts, etc., it will be a better article.
Re: Reviewer 1 and Reviewer 2 gave very similar comments. The authors agree that your comments should be improved and are the weakest point of this paper. However, evaluating the environmental impact of new materials is a whole new research. In this study, the authors are discussing the civil-military's new facility policy in terms of sustainability.
Round 2
Reviewer 1 Report
The authors addressed this reviewer's comments well, while improving the quality of the manuscript.